# Novel Administration Routes, Delivery Vectors, and Application of Vaccines Based on Biotechnologies: A Review

**DOI:** 10.3390/vaccines12091002

**Published:** 2024-09-01

**Authors:** Chung-I Rai, Tsu-Hsiang Kuo, Yuan-Chuan Chen

**Affiliations:** 1Department of Cosmetic Science, Vanung University, 1, Van Nung Road, Chung-Li City 320676, Taiwan; barry.rai@gmail.com; 2Department of Rehabilitation Science, Jenteh Junior College of Medicine, Nursing and Management, Miaoli County 356006, Taiwan; dennissylvie@gmail.com; 3Department of Biotechnology and Pharmaceutical Management, Jenteh Junior College of Medicine, Nursing and Management, Miaoli County 356006, Taiwan; 4Department of Nursing, Jenteh Junior College of Medicine, Nursing and Management, Miaoli County 356006, Taiwan; 5Department of Medical Technology, Jenteh Junior College of Medicine, Nursing and Management, Miaoli County 356006, Taiwan; 6Program in Comparative Biochemistry, University of California, Berkeley, CA 94720, USA

**Keywords:** vaccine, biotechnology, vector, nanoparticle, recombination DNA, mRNA, therapeutics, adjuvant, delivery system

## Abstract

Traditional vaccines can be classified into inactivated vaccines, live attenuated vaccines, and subunit vaccines given orally or via intramuscular (IM) injection or subcutaneous (SC) injection for the prevention of infectious diseases. Recently, recombinant protein vaccines, DNA vaccines, mRNA vaccines, and multiple/alternative administering route vaccines (e.g., microneedle or inhalation) have been developed to make vaccines more secure, effective, tolerable, and universal for the public. In addition to preventing infectious diseases, novel vaccines have currently been developed or are being developed to prevent or cure noninfectious diseases, including cancer. These vaccine platforms have been developed using various biotechnologies such as viral vectors, nanoparticles, mRNA, recombination DNA, subunit, novel adjuvants, and other vaccine delivery systems. In this review, we will explore the development of novel vaccines applying biotechnologies, such as vaccines based on novel administration routes, vaccines based on novel vectors, including viruses and nanoparticles, vaccines applied for cancer prevention, and therapeutic vaccines.

## 1. Introduction

Vaccines are biological preparations made of pathogenic microorganisms to induce specific adaptive immune responses to microbial infection by, e.g., bacteria, viruses, etc. Vaccination is one of the best strategies for protecting uninfected people, reducing moderate–severe symptoms and even death, and decreasing the spreading rate and scope of infectious diseases. Many countries have tried to increase the rate of vaccination to reduce the transmission, mortality, and morbidity of infectious diseases. However, some factors may hinder vaccination promotion, such as the limitations of the adaptive age population, breakthrough infection, and safety concerns [1]. Additionally, the side effects, expense, discomfort, inconvenience, and the need for professional assistance during vaccination may cause vaccine hesitancy. Therefore, scientists have been focusing on the development of new methods to overcome these disadvantageous issues for vaccine promotion.

Recently, there has been progress in biotechnologies for vaccine production, including the development of microneedles for minimal invasion, dry powder for inhalation, novel vectors (e.g., virus, nanoparticle) for efficient delivery, and the addition of adjuvants for efficacy boost. The advancement of biotechnologies has made it possible to provide multiple/alternative administration routes for the improvement of the safety, effectiveness, convenience, and tolerance of vaccines. Traditional vaccines are only used for the prevention of infectious diseases, while novel vaccines can also be used for the prevention and therapy of noninfectious diseases, especially for cancer. Some novel vaccines administered through microneedles or inhalation and therapeutic cancer vaccines based on biotechnologies have been approved for the market or are being developed.

## 2. Vaccination through Novel Administration Routes

Traditional vaccines can mainly be classified into inactivated vaccines, live attenuated vaccines, and subunit vaccines given orally or via intramuscular (IM) injection or subcutaneous (SC) injection. Traditional vaccine delivery has widely known advantages, but these methods of administration have some restrictions or shortcomings, such as pain, anxiety, risk of infection (e.g., needle stick injuries or needle reuse), or the requirement of cold-chain and professional involvement, etc. (Table 1). Consequently, novel vaccines administered by multiple/alternative routes using biotechnologies, including microneedle and inhalation, have been developed to make vaccines more secure, effective, tolerable, and universal in resource-limited areas, alleviating the disadvantages, risk, or limitations of traditional administration routes (Table 1).

### 2.1. Microneedle

The microneedle array (MNA) is fabricated in different shapes, types, configurations, and formats and made of different materials. A microneedle patch (MNP), usually composed of a small array containing several to hundreds of microneedles, is considered as a device for transdermal drug delivery. The microneedles are arranged on the small patch to become a hypodermic needle and a transdermal patch; they are applied to the patient’s skin by physically puncturing the stratum corneum for a period of time to effectively deliver the drugs (Figure 1). Microneedle percutaneous immunization is obtained by penetrating the stratum corneum to reach the epidermis so that the vaccine is easily and rapidly recognized by epidermal antigen-presenting dendritic cells (e.g., Langerhan cells) to induce specific immune responses, especially for adaptive immunity [4]. Because the vaccine delivery via microneedles is painless, self-administered, and can produce superior immunogenicity rapidly, it has been seen as a potential alternative route to traditional delivery. The recent emergence of the cleanroom-free, 3D, or additive manufacturing of microneedles will reduce manufacturing costs and time significantly, making this method commercially attractive [19,20,21]. Vaccine delivery through the Nanopatch^®^ system has shown excellent efficacy, dose-sparing capacity, and stability [19,20,21]. Additionally, the microneedle only needs single-dose administration systems and does not require reconstitution or cold-chain storage [21]. The other characteristics of microneedles include ease of self-administration, pain-free, high efficacy, minimal invasion, and economic advantages. These advantages may enable the global distribution of vaccines and result in higher patient compliance.

The developing microneedle vaccines are described and illustrated below.

#### 2.1.1. Severe Acute Respiratory Syndrome Coronavirus 2 (SARS-CoV-2)

In 2024, Vander Straeten et al. manufactured a thermostable mRNA vaccine in a MNP format [22]. They reported an automated process for printing MNP coronavirus disease 2019 (COVID-19) mRNA vaccines in a dependent device. The vaccine ink consisted of mRNA encapsulated in a lipid nanoparticle, and a dissolvable polymer mixture was optimized by screening formulations in vitro. The results showed that MNPs were shelf-stable at room temperature for at least 6 months [22]. This mRNA vaccine was proven to be efficacious and could be delivered with a single patch via loading efficiency and microneedle dissolution tests. In animal studies, mice generated long-term immune responses using MNPs with mRNA encoding SARS-CoV-2 spike protein receptor-binding domain (RBD), similar to those of IM injection [22]. This study suggests that the mRNA microneedle vaccine automatically printed was potentially effective against COVID-19 without the requirement of cold-chain storage and trained healthcare personnels. This is advantageous for vaccination promotion in communities without enough equipment and professionals.

#### 2.1.2. Influenza Virus

In 2024, Kang et al. developed an influenza vaccine loaded with dissolvable egg microneedles and characterized the specificity of layer-specific functions by distinguishing formulations [23]. They designed the formulations of dissolving microneedles specific to the mechanical properties of the microneedle so that the formulation could maintain the activity of the vaccine and had mechanical strength. They quantitively evaluated the antigen activity of the formulation candidates using an enzyme-linked immunosorbent assay (ELISA) and a single radial immunodiffusion assay (SRIDA) in vitro and found their activity to be 87% and 91%, respectively [23]. Furthermore, they conducted in vivo tests in mice inoculated with the formulation constructed into egg microneedles to determine the protective efficacy against influenza virus [23]. This study reveals that these egg-based dissolving microneedles with functionalized formulations can provide protective immune responses equal to a fractional dose via IM injection [23].

#### 2.1.3. Japanese Encephalitis Virus

In 2022, Iwata et al. evaluated the safety and dose-sparing effect of an MNP against Japanese encephalitis in healthy volunteers aged 20~34 years who had never received vaccines and were never naturally infected with Japanese encephalitis virus [24]. In total, 39 eligible participants were enrolled and randomly assigned to three groups (*n* = 13 per group) in a phase-I clinical trial between 31 August and 2 September, 2019. The three groups receiving inactivated Japanese encephalitis vaccines were administered twice, 3 weeks apart, as follows: (1) 2.5 μg/injection via SC injection; (2) 0.63 μg/patch via high-dose MNP (25%); (3) 0·25 μg/patch via low-dose MNP (10%) [24]. They found that all participants in the MNP groups only had a local erythematous reaction, and no serious adverse events were observed. The amount of vaccine delivered via MNP to each participant was 0.63~1.15 μg (50–92%) of the full 1.26 μg for the MNA (25%) group and 0.25~0.41 μg (51–84%) of the full 0.50 μg for the MNA (10%) group, respectively [24]. Additionally, at day 42 after first immunization, the mean titers (log_10_) of the neutralizing antibody were 2.55 for MNA (25%), 2.04 for MNA (10%), and 2.08 for SC injection, respectively [24]. Although the total dose of the two MNP groups were lower than the SC group, the high-dose MNP group still produced more neutralizing antibodies, while the low-dose MNP group produced approximately the same amount as the SC group. This study shows that the MNP can be a safe, tolerable, effective, and dose-sparing vaccine, suggesting a possible means to to reduce the necessary number of immunogens [24].

#### 2.1.4. Human Papillomavirus (HPV)

In 2022, Ray et al. developed a dissolvable microneedle loaded with a thermally stable HPV vaccine candidate composed of Qβ virus-like particles (VLPs) displaying a highly conserved epitope from the L2 protein of HPV (Qβ-HPV) to improve vaccination distribution and administration [25]. They found that microneedle delivery of Qβ-HPV with a smaller amount of intradermal dose produced similar amounts of anti-HPV16 L2 IgG antibodies compared to SC injection [25]. The vaccine candidate was stable at room temperature for several months and yielded neutralizing antibodies after immunization [25]. The Qβ VLP and its delivery technology could serve as a universal platform with a wide range of applications, like a plug-and-play system [25]. This study suggests that this system, combining dissolvable microneedles with VLP, provides the possibility for the development of novel HPV vaccines, which are low cost, stable at room temperature, dose-sparing, and accessible to the majority of global population [25].

#### 2.1.5. Clostridium Botulinum

In 2022, Zhao et al. developed a thermal-stable, dissolving MNP to deliver a protein vaccine using a recombinant C-terminal heavy chain of botulinum neurotoxin serotype A (BoNT/A) (Hc of BoNT/A, AHc) to prevent botulism [26]. They optioned fish gelatin (a natural non-toxic and bacteriostatic material) as the microneedle matrix to prepare a dissolving microneedle vaccine. They found that the fish gelatin matrix, at high concentrations, had excellent bacteriostatic properties, mechanical performance, and vaccination effects [26]. The microneedle vaccine containing different antigen doses protected the mice against 10^6^ LD_50_ of BoNT/A injected intraperitoneally, showing the same efficacy as SC injection [18]. The AHc vaccine did not denature after 7 days of storage at 37 °C and kept good immunogenicity and protective efficacy after 6 months of storage at room temperature [26]. This study demonstrates that an AHc vaccine without cold-chain storage can be successfully prepared using a bacteriostatic MNP containing fish gelatin matrix, indicating the possibility of effective, convenient, painless, and large-scale vaccination [26].

### 2.2. Inhalation

The increasing concerns of pandemic respiratory pathogens highlight the importance of developing systemically administered vaccines to induce respiratory mucosal immunity to respiratory pathogens such as SARS-CoV-2 [27,28], influenza virus [29,30], respiratory syncytial virus (RSV) [27,31], and *Mycobacterium tuberculosis* [32], etc. Inhalable vaccines promise to trigger unique immune responses in the linings of the airways, i.e., mucosal immunity. They can render more immediate protection against any pathogens entering the body via the nose and mouth. The inhalation of vaccines may induce immune responses that are more similar to the responses following a natural infection of respiratory pathogens. Different from traditional vaccines, inhalable vaccines are able to prime local IgA-mediated immune responses, which prevent future infections via neutralizing dimeric IgA that are secreted into the airway lumen [8]. This kind of vaccine provides triple protection, including humoral immunity, cellular immunity, and mucosal immunity with tissue resident memory T and B cells. It has one more type of immunity than systemic immunity induced via traditional injections, namely, mucosal immunity, which targets the mucosal infection site for improving vaccine efficacy.

Inhalation is an efficient way to target the respiratory tract to induce immune responses. Inhalable vaccines, currently divided into nasal spray vaccines and oral inhalable vaccines, are potentially successful in achieving effective and robust immunization via safe, easy, painless, and affordable methods if they are consistent with the following principles [28]: (1) the vaccine is stable at room temperature, not requiring a cold-chain for storage and not requiring sterile water for reconstitution; (2) the vaccine should be delivered to the right site in the lungs; (3) the vaccine should be active enough to overcome the barrier of immunotolerance, ensuring the activation of sufficient mucosal immune cells; (4) the vaccine should not result in serious-but-rare adverse reactions such as asthma and chronic obstructive pulmonary disease (COPD) [28].

Nasal spray vaccines containing specific formulations can either inactivate respiratory pathogens or block their entry into cells to decrease viral loads in the nose, thus preventing the virus from spreading to the lung or to neighboring people. Nasal mucosa vaccination using disposable inhalers will elicit the release of mucosal IgA and serum IgG rapidly and effectively. Moreover, it is able to enhance cross-protection via the production of cross-reactive antibodies [33]. Nasal spray vaccines can cause antibodies to accumulate in the upper respiratory tract, but they have limited protection against low respiratory tract infections. Therefore, nasal spray vaccination is suitably applied for upper respiratory infections via aerosol or the air. Most of the available nasal spray vaccines are low-cost, amenable for self-administration, and do not require cold-chain storage, healthcare staff, or a facility [33].

Unlike nasal spray vaccines, oral inhalable vaccines are taken through the mouth as a fine mist administered via a nebulizer. The large droplets would prime the mouth and throat, while the small droplets would travel further into the body [34]. Nasal sprays mainly reach the nose and throat, while oral inhaled aerosols can bypass the nasal passage and deliver vaccine droplets deep in the low respiratory tract or gastrointestinal tracts, where they can induce broad protective immune responses. They will cause antibodies to accumulate in the entire airway and gastrointestinal tracts, providing a stable immune response including humoral and cellular immunity. Infections in the upper respiratory tract are usually not severe, but infections in the low respiratory tract often lead to serious illness and even death. Nasal spray vaccines have been shown to be highly effective in children, but much less effective in adults [34,35]. Therefore, oral inhalable vaccines are a more suitable choice for adults than nasal spray vaccines.

The developing inhalable vaccines are described and illustrated below.

#### 2.2.1. SARS-CoV-2

In 2023, Ye et al. developed an inhalable, single-dose, dry-powder aerosol SARS-CoV-2 vaccine that induces systemic and mucosal immune responses [36]. The vaccine encapsulates assembled nanoparticles containing cholera toxin B subunits presenting the SARS-CoV-2 RBD antigen. In microcapsules with optimal aerodynamic size, the nano–micro coupled structure could efficiently facilitate alveoli delivery, sustained antigen release, and antigen-presenting cell uptake. The results suggested that this vaccine stimulated mass production of IgG, IgA, and immune responses of T cells, providing effective protection against SARS-CoV-2 in mice, hamsters, and nonhuman primates [36]. Importantly, this vaccine could induce broad immunity via inhalation the mouth via a nebulizer [36]. They also revealed a vaccine which co-displays ancestral and Omicron antigens, which could extend antibody responses to co-circulating strains and inhibit Omicron variant transmission [36]. Moreover, the new inhalable dry-powder vaccine did not require cold-chain storage and could be stored at room temperature for one month [36]. These findings suggest that the oral inhalable vaccine with the specific nano–micro coupled structure is as a promising platform against COVID-19 and other infectious respiratory diseases.

In 2023, Elder et al. used an inhaled subunit vaccine candidate (ISR52) based on the SARS-CoV-2 spike S1 protein to induce local mucosal immune responses [37]. In lethal challenge hACE2 transgenic SARS-CoV-2 mice, they found that intranasal and intratracheal administration of ISR52 provided better protection against severe infection than the SC injection of the vaccine [37]. The results demonstrated that inhaled ISR52 stimulated both CD4 and CD8 T-cell spike-specific responses, and the responses were maintained for at least 6 months in wild-type mice [37]. The induced IgG and IgA responses cross-reacted with several SARS-CoV-2 variants of concern in the lung and serum. Moreover, the protected animals were found to produce neutralizing antibodies against viruses. This study shows that it is possible to develop ISR52 as a dry powder formulation for inhalation that does not require cold-chain storage (though ISR52 needs to be further evaluated in a phase-I/II clinical trial) [37].

#### 2.2.2. Influenza Virus

In 2021, Jeong et al. developed a nasal nanoparticulate vaccine (NanoVac) which contained a new self-assembled immunization system pertaining to a photoactivatable polymeric adjuvant with the hemagglutinin of the influenza virus to demonstrate photochemical immunomodulation [38]. In animal studies, they found that NanoVac could increase the retention period of antigens via spatiotemporal photochemical modulation in the nasal cavity to avoid fast removal when the vaccines pass through nasal mucosa and mucociliary clearance [38]. The results revealed that photochemical immunomodulation of NanoVacs successfully induced humoral and cellular immune responses after the immune cells were stimulated, leading to the secretion of specific antibodies, cytokines, and CD8^+^ T cells in mice [38]. Furthermore, the mice showed significant prevention from viral infection, followed by challenging with the influenza virus after nasal immunization via NanoVacs [38]. This study suggests that the nanoparticle-based nasal vaccine (NanoVac) shows promise in the prevention influenza, which is currently active and prevalent.

#### 2.2.3. Respiratory Syncytial Virus (RSV)

In 2024, Eberlein et al. developed a noninvasive mucosal vaccination via inhalation to offer an alternative to RSV, potentially resulting in efficient and fast elimination after viral natural infection [39]. They applied low-energy electron-inactivated RSV (LEEI-RSV), formulated with phosphatidylcholine-liposomes (PC-LEEI-RSV) or 1,2-dioleoyl-3-trimethylammonium-propane and 1,2-dioleoyl-sn-glycero-3-phosphoethanolamine (DD-LEEI-RSV), for the intranasal vaccination of mice [28]. LEEI-RSV and formalin-inactivated-RSV (FI-RSV) were used via IM vaccination as the controls. The authors detected the presence of RSV-specific IgA antibodies and a significant reduction in viral load upon challenge in mucosal DD-LEEI-RSV-vaccinated animals [39]. The results demonstrate that Alhydrogel-adjuvanted LEEI-RSV via IM had a Th2-bias, with enhanced IgE, eosinophils, and lung histopathology comparable to FI-RSV. However, these effects were absent when applying the mucosal vaccines. This study suggests that the mucosal vaccine formulated with DD-LEEI-RSV is promising for the prevention of RSV infection [39].

#### 2.2.4. Mycobacterium Tuberculosis

In 2022, Gomex et al. developed a vaccine candidate, ID93 + GLA-SE, which is a thermostable dry-powder vaccine against tuberculosis [40]. This dry powder was advantageous in that it reduced the need for cold-chain storage and provided flexibility in dosage and administration routes. The authors evaluated the immunogenicity and protective efficacy of spray-dried ID93 + GLA-SE in a murine model compared with relevant controls [40]. The four different administration routes for the spray-dried ID93 + GLA-SE were as follows: (1) reconstitution and IM injection; (2) reconstitution and intranasal delivery; (3) nasal dry powder delivery via inhalation; and (4) pulmonary dry powder delivery via inhalation [40]. Optimization using representative vaccine-free powder revealed that about 10% and 44% maximum dose would be delivered for intranasal delivery and pulmonary delivery, respectively [29]. The results showed that both intranasal reconstituted vaccine delivery and pulmonary dry powder vaccine delivery can control the growth of *Mycobacterium tuberculosis* in infected mice, comparable to IM delivery [40]. These two vaccinated groups showed improved protection in the presence of cytokine-producing T cell responses compared with control groups. This study suggests that a novel vaccine, whose formulation is dry powder and whose delivery is via nasal inhalation, can potentially offer protection against *Mycobacterium tuberculosis* infection [40].

## 3. Vaccines Based on Novel Vectors

### 3.1. Viruses

Virus expression vectors are genetically engineered to carry an expression cassette containing the desired gene or to have deficiency in specific genes. They are structurally similar to the wild-type virus, infectious, and still can replicate in hosts, but they are usually nonpathogenic. The viral expression vector vaccine is employed in the use of a virus to deliver genetic material (DNA) to encode specific antigens to induce immune responses. It can enable antigen expression within cells and trigger both humoral and cellular immune responses, unlike subunit vaccines, which only induce humoral immunity. Most virus expression vectors used for vaccine development are lacking toxic genes and deficient in the genes associated with replication [41,42]. The safety of vaccines is the major concern with respect to their wide acceptance and approval for medical use; therefore, the virus expression vector vaccines require a high biological safety level, and non- or low-pathogenic viruses are priority choice. Currently, several viruses have been designed as vaccine vectors, such as retrovirus, Sendai virus, lentiviruses, vaccinia virus, adenovirus, adeno-associated virus (AAV), cytomegalovirus, and vesicular stomatitis virus (VSV) [41]. The adenovirus is the most commonly used virus expression vector because it can elicit robust immune responses. For example, COVID-19 vaccines and Ebola virus vaccines using virus expression vectors have been authorized for clinical application (Table 2).

The developing vaccines delivered via virus expression vectors are described and illustrated below.

#### 3.1.1. Zika Virus

In 2020, Bullard et al. developed a Zika virus vaccine using a human adenovirus type 4 (Ad4-prM-E) and an Ad5 vector (Ad5-prM-E), respectively [50]. They observed that vaccination with Ad4-prM-E resulted in a strong T cell response without inducing antibodies against Zika virus, while vaccination with Ad5-prM-E resulted in both humoral immunity and T-cell responses in C57BL/6 transgenic mice [50]. However, both vectors provided anti-Zika activity in a lethal mice model challenged with Zika viruses. They also confirmed that the T-cell biased immune response is not mouse-strain-specific in that vaccination of BALB/c mice led to immune responses like C57BL/6 mice [50]. The mice vaccinated with an Ad4-expressing influenza hemagglutinin (HA) presented T-cell responses against HA without the significant production of anti-HA antibodies, suggesting that these T-cell response are unique to the Ad4 serotype, rather than the transgenic expression [50]. The authors found that the simultaneous administration of a UV-inactivated Ad4 vector with the Ad5-prM-E would cause a significant reduction in antibody production. The results showed that this serotype-specific immune profile was induced by different adenovirus vector types, and it was important to characterize these alternative Ad serotypes continually [50]. This study shows that a vaccine containing influenza HA and Ad5-prM-E is promising against Zika virus infection.

#### 3.1.2. Mycobacterium Tuberculosis

In 2022, Jeyanathan et al. evaluated the safety and immunogenicity of a tuberculosis vaccine using the human serotype-5 adenovirus-based vector (AdHu5Ag85A) delivered to humans via inhaled aerosol or IM injection, respectively [51]. In total, 31 healthy adults who had ever received BCG vaccine were enrolled in the phase-Ib clinical trial. The participants were divided into three groups, including the low-dose (LD) and high-dose (HD) of AdHu5Ag85A aerosol administration groups and the IM injection group [51]. The adverse events were assessed at various times after vaccination. The results revealed that both LD and HD, given via aerosol inhalation and IM injection, were safe and well tolerable. Both aerosol doses, particularly LD vaccination, significantly promoted the production memory CD4^+^ and CD8^+^ T cells in airway tissues compared to IM injection [51]. They found the LD aerosol vaccination induced Ag85A-specific T cell responses in the blood, like the IM injection. They concluded that inhaled aerosol delivery of the Ad-vectored vaccine conferred a safe and superior means of triggering mucosal immunity in the respiratory tract [51]. This study suggests that an inhalable aerosol vaccine delivered by an adenovirus vector can potentially protect the infection of respiratory pathogens, including *Mycobacterium tuberculosis* and SARS-CoV-2.

#### 3.1.3. Plasmodium

In 2017, Milicic et al. studied the protective efficacy of a single-dose adenovirus-vectored malaria vaccine with compound adjuvants in mice which had malaria [52]. Of the adjuvants tested, they found that only Abisco^®^-100 and CoVaccineHT^TM^ could enhance vaccine efficacy and protection from pre-erythrocytic malaria, while mice were challenged with pathogens [52]. Both of these adjuvants induced CD8^+^ T cells to express the CD107a marker without the function of IFNγ, TNFα, and IL2. However, the CoVaccineHT^TM^ adjuvanted vaccine induced more antigen-specific central memory CD8^+^ cells than Abisco^®^-100 [52]. The results showed that the efficacy of an adenovirus-vectored malaria vaccine could be enhanced using specific adjuvants, and their co-administration was able to provide full protection via single-dose immunization [52]. Some adjuvants were demonstrated to improve the existing protective efficacy of viral vectored vaccines by increasing CD8^+^ T cell immune responses. This study suggests that a single-dose adenovirus-vectored vaccine with adjuvants is promising for the prevention of malaria.

### 3.2. Nanoparticles

Nanoparticles–the typical size ranges from 20 to 100 nanometers (nm)—are usually made from gold, carbon, dendrimers, polymers, or liposomes. At present, nanoparticles have been found that are capable of activating immune responses and cytokine production. Therefore, they have the potential to be used as delivery vehicles and adjuvants to enhance immune responses for vaccine development. Current nanoparticles applied for enhancing immunogenicity are referred to as nano-immuno-activators or stimulators, including inorganic nanoparticles like iron, aluminum, clay and silica, polymeric nanoparticle like poly(d,l-lactide-co-glycolide) (PLG), poly (d,l-lactic-coglycolic acid) (PLGA), poly (g-glutamic acid) (g-PGA), poly (ethylene glycol) (PEG), immunostimulating complex, liposomes, self-assembly proteins, emulsions, and VLPs [53,54]. By encapsulating antigenic materials, nanoparticles are able to protect antigens and adjuvants from premature degradation. The nanoparticle-based delivery system potentially extends the duration of antigen stability, enhances the uptake of antigens, activates dendritic cells, and promotes cross-presentation [53,54]. The ease of use and multiple functions of nanoparticles have made them a practical and efficient delivery tool for vaccine candidates. For example, several COVID-19 mRNA vaccines and influenza vaccines prepared via nanoparticles have been authorized for clinical application (Table 3).

Because the size scale of the virus is the nanometer, viruses have been considered to be naturally occurring nanoparticles. They can be used as prefabricated virus nanoparticles for gene delivery, vaccine development, or immunotherapy intervention, such as mammalian viruses (e.g., AAV), plant viruses (e.g., Cowpea mosaic virus), and bacteriophages. Virus nanoparticles are capable of replicating and eliciting both innate and adaptive immune responses in hosts (Table 4).

VLPs, specialized nanoparticles, demonstrate self-assembling capacity and carry many traits mimic to viruses but lose the ability to replicate in hosts as they are lacking a viral genome. Because of the immunological characteristics of VLPs, including size, structure, component, and repetitive surface geometry, they can induce both innate and adaptive immune responses, especially robust antibody responses [59,60,61]. Consequently, VLP-based vaccines present the parameters of most traditional vaccines, rendering a safe platform and being economical for vaccine development [59,60,61] (Table 4). Several VLP-based vaccines are commercially available, such as HPV vaccines (e.g., Cervarix^®^, Gardasil^®^, Gardasil 9^®^), and the second and third-generation HBV vaccines (ENGERIX-B, Sci-B-Vac™, respectively) [59].

The developing vaccines based on nanoparticles are described and illustrated below.

#### 3.2.1. Influenza Virus

In 2024, Hardenberg et al. explored whether polymeric nanoparticles can efficiently deliver influenza hemagglutinin mRNA to elicit protective immune responses in ferrets [62]. The lipid nanoparticles they used was composed of biodegradable poly (amido) amine-based polymers with mRNA payload like SARS-CoV-2 mRNA vaccines. In this animal model, they found that the nanoparticle-based vaccine induced strong humoral and cellular immune responses without local and systemic immunoreactivity [62]. They found that vaccinated animals showed decreased clinical signs, symptoms, and viral load once they were challenged with influenza virus compared with control ones [62]. However, it is necessary to investigate the polymeric nanoparticles in the context of vaccines. Future studies will focus on the optimization of the payload, the stability of nanoparticles, and the efficacy of pre-existing immunity. The results show that the mRNA vaccine delivered by nanoparticles is promising for the prevention of influenza [62].

In 2023, Pan et al. reported an intranasal vaccine based on multivalent epitope nanoparticles to confer broad protection against influenza A and B viruses [63]. They constructed three highly conserved epitopes composed of the A α-helix of hemagglutinin (H), the ectodomain of matrix protein 2 (M), and the HCA-2 of neuraminidase (N) to present a self-assembling recombinant human heavy chain ferritin cage (F) to produce HMNF nanoparticles [63]. In the mouse model, they found that the HMNF-nanoparticle-based intranasal vaccine induced specific antibody production and T-cell-mediated immunity. These strong immune responses showed cross-reactivity to various antigen mutations [63]. The results revealed that vaccination with HMNF nanoparticles provide the mice with full protection against divergent influenza A and B viruses due to the synergistic effects of antibodies and T cells. Additionally, the immune responses elicited by intranasal vaccine with HMNF nanoparticles could be maintained six months after vaccination [63]. This study suggests that the HMNF-nanoparticle-based intranasal vaccine has potential against influenza.

#### 3.2.2. Epstein–Barr Virus (EBV)

In 2023, Sun et al. developed a gB nanoparticle (gB-I53-50 NP) to show multiple copies of gB, which is an EBV fusion protein that mediates host cell recognition and membrane fusion entry, making it critical for viral infection in B cells and epithelial cells [64]. In the mouse and non-human primate preclinical model, gB-I53-50 NP shows improved structural integrity and stability to enhance immunogenicity compared with the gB trimer [64]. The results showed that vaccination with gB-I53-50 NP induced a robust and persistent antibody response that protected mice against lethal EBV challenge [64]. This study shows that the nanoparticle-based vaccine candidate containing gB fusion proteins potentially provides a platform for the prevention of EBV infection, which currently has no available prophylactic vaccine.

#### 3.2.3. *Borrelia burgdorferi*

In 2023, Pan et al. utilized the lipid-nanoparticle-encapsulated nucleoside-modified mRNA (mRNA-LNP) platform to produce a vaccine against Lyme disease, similar to COVID-19 mRNA vaccines [65]. The pathogenic agent of Lyme disease, outer surface protein A (OspA) expressed by *Borrelia burgdorferi*, is the most promising antigen candidate for vaccine development [65]. They developed a OspA-encoding mRNA-LNP vaccine and assessed its immunogenicity and protective efficacy compared to an alum-adjuvanted OspA protein subunit vaccine. The results demonstrated that OspA mRNA-LNP elicited better humoral and cellular immunity in mice followed by a single immunization [65]. Their study revealed that efficient mRNA vaccines can target specific antigens, resulting in protection against bacterial infection [65]. This study suggests that the nanoparticle-based mRNA vaccine candidate offers a possible opportunity for the prevention of Lyme disease, for which a prophylactic vaccine is still unavailable.

#### 3.2.4. Respiratory Syncytial Virus (RSV)

In 2019, Marcandalli et al. evaluated whether the prefusion structure of the RSV fusion (F) glycoprotein and its identification could be the target of neutralizing antibodies to develop an effective vaccine [66]. They found that a self-assembling nanoparticle could display a prefusion-stabilized variant of the glycoprotein trimer (DS-Cav1) in a repetitive array on its surface [66]. Highly ordered and monodisperse DS-Cav1 immunogens were induced for generation according to the two-component nature of the nanoparticle structure. In the mice and nonhuman primates model, the results showed that the nanoparticle immunogen presenting 20 DS-Cav1 trimers induced more neutralizing antibodies than trimeric DS-Cav1 by about 10-fold [66]. This study suggests that the RSV vaccine candidate containing F glyproteins and the designation of two-component nanoparticles via computation can work as a platform for scaffold-based vaccines against RSV.

Currently, several vaccines based on viral vectors or nanoparticles have been successfully utilized for clinical application, and some of them are being developed. These vectors seem to be promising for the development novel vaccines. Viruses and nanoparticles have many advantages and versatile applications, but it necessary to overcome and avoid some disadvantages, limitations, and risks (Table 5).

## 4. Vaccines Used for Cancer Prevention

The causes of cancers include environmental (e.g., tobacco, diet, pollution), physical (e.g., radiation), chemical (e.g., carcinogens), and biological factors (e.g., microorganism). Infections with specific viruses, bacteria, and parasites are biological factors causing about 16–18% of cancers worldwide [74]. These infectious agents include *Helicobacter pylori*, hepatitis B virus (HBV), hepatitis C virus (HCV), human papillomavirus (HPV), Epstein–Barr virus (EBV), human T-lymphotropic virus 1 (HTLV-1), and Kaposi’s sarcoma-associated herpesvirus (KSHV), etc. [75].

Vaccines are conventionally considered to be prophylactic agents for infectious diseases, not for noninfectious diseases such as neurodegenerative disorders, metabolic diseases, cardiovascular diseases, and cancers, etc. However, it is expected that some specific cancers can be prevented if their etiologic factor is infectious agents. Prophylactic vaccines against cancer provide an opportunity to protect the human from the risk of tumor development. Two important issues related to prophylactic cancer vaccines are the identification of suitable populations for vaccination and the establishment of solid tumor-specific immunologic memory [76]. The overexpressed or mutated proteins and growth factor receptors, as tumor-associated antigens, can be used to produce potential targets for specific immunoprevention [77].

Currently, vaccines have been developed and approved to prevent cancer caused by some oncoviruses, such as HPV and HBV [78]. HPV vaccines can reduce the risk of cervical cancer [79,80], and HBV vaccines can reduce the risk of hepatocellular carcinoma (HCC) [81,82]. For example: Gardasil^®^ (4vHPV) was the first HPV 4-valent vaccine to be approved by the US Food and Drug Administration (FDA) for females aged 9~26 to prevent cervical cancer, genital warts, and vulvar and vaginal precancerous lesions in 2006 [83]. It was reported that a prophylactic vaccine that could confer durable protection against HPV infection and related diseases [84]. Gardasil-9 ^®^ (9vHPV), the second generation of HPV, a 9-valent vaccine approved by the US FDA for females aged 13~26 and males aged 13~21 in 2014, has been indicated to protect against five additional high-risk HPV types (31, 33, 45, 52, and 58) [85,86]. Additionally, HBV vaccination has been attested to decrease, definitely, the prevalence of chronic HBV infection, to offer significant effects for the prevention of HCC, and to reduce mortality associated with HCC and liver diseases [87,88].

## 5. Vaccines Used for Pathogen Therapeutics

The rabies vaccine is used to help prevent a person from suffering rabies symptoms if there is a chance he has been exposed to rabies viruses. The rabies virus, a single-stranded RNA virus belonging to Rhabdoviridae family, has a very special spreading route in the human body. After the rabies virus infection, it first replicates in skeletal muscle cells near the wound. The viral G protein spike can interact with the nicotinic acetylcholine receptor on the cell membrane to form endophagic vesicles and enters the cytoplasm [89,90]. After replication, the virions are released from the skeletal muscle cell to passes through the motor end plate at the neuromuscular junction and enter the axon at a speed of 5–100 mm per day [89,90]. Therefore, it will take a lot of time for the rabies virus to invade the central nervous system (brain and spinal cord) to cause 100% morbidity via retrograde axonal transport, not blood flow. Premorbid treatment can include rabies vaccination, but the best time is within 24 h after being bitten by animals with rabies viruses [80,81]. Because the vaccine induces B lymphocytes to produce antibodies faster than the rabies viruses reach the central nervous system, vaccination after the bitten is still effective even though the virus has invaded into body [91,92]. For human beings, rabies is usually preventable with postexposure vaccination using inactivated vaccines (e.g., a human diploid cell vaccine) before the onset of diseases, different from animals, for whom pre-exposure vaccination for prophylaxis using live attenuated vaccines is necessary. Consequently, the rabies vaccine is considered to be a therapeutic vaccine.

## 6. Vaccines Used for Cancer Therapeutics

Cancer immunotherapy can be divided into two types: passive type and active type. The former is intended to provide antibodies or adoptive T cells to destroy tumor cells; the latter is intended to administer vaccines or cytokines to treat cancer by stimulating or restore immune responses. Cancer vaccines can be further classified into preventive vaccines and therapeutic vaccines based on their specific mechanism. The preventive vaccine is used for the prevention of cancer caused by oncoviruses like HPV and HBV. The therapeutic vaccine is used for the treatment of existing cancer; therefore, it changes both in target antigens and vaccine platforms, different from prophylactic vaccines. Currently, three therapeutic cancer vaccines have been approved by the US FDA: TICE^®^ (intravesical bacillus Calmette-Guerin (BCG) live) for bladder cancer in 1998; Provenge^®^ (sipuleucel-T) for prostate cancer in 2010; and Imlygic™ (T-VEC) for metastatic melanoma in 2015, respectively [93]. Previous clinical trials have revealed that therapeutic vaccines usually show excellent tolerance, as well as the ability to target tumor neoantigens and induce antigen cascade [93]. Ongoing studies are focusing on improving vaccine efficacy either by targeting neoantigens of tumor cells, combining vaccines with nanoparticles or adjuvants, or by linking vaccines with standard therapies or other immunotherapies [93].

### 6.1. Metastatic Melanoma Vaccine

In 2024, D’Alise et al. used a vector-based personalized vaccine NOUS-PEV in combination with pembrolizumab for the treatment of patients with metastatic melanoma who had never received any treatment in a phase-Ib clinical trial [94]. NOUS-PEV consists of priming with a nonhuman great ape adenoviral vector (GAd20) and is capable of expressing 60 neoantigens. The authors evaluated all evaluable patients receiving the prime/boost regimen and detected robust neoantigen-specific immune responses, including both CD4 and CD8 T cells, to multiple neoantigens [94]. The results showed that NOUS-PEV can improve the capacity of tumor-reactive T cells to enable variable, robust, and persistent antitumor immune responses [94]. This study suggests that personalized vaccines, which target multiple neoantigens, are promising in inducing diverse T-cell responses to tumor heterogeneity.

### 6.2. HPV-Related Malignancy

In 2024, Wang et al., developed an mRNA-based HPV therapeutic vaccine mHTV-02 formulated by lipid nanoparticles to target the E6/E7 genes of HPV16 and HPV-18 in animals [95]. They found that mHTV-02 significantly elicited antigen-specific cellular immunity and potent memory T-cell immunity [95]. Additionally, the significant CD8^+^ T-cell infiltration and cytotoxicity in TC-1 tumors expressing HPV E6/E7 led to tumor suppression and improved mice survival. The intramuscular or intratumoral injection of mHTV-02 also demonstrated better therapeutic effects in reducing tumor size or prolonging mice survival compared with vaccine delivery via IV injection [95]. This study suggests that mHTV-02 via specific administration routes could be a therapeutic mRNA vaccine candidate for the treatment of malignancies related to HPV16 or HPV18 infections.

### 6.3. Liver Cancer

In 2023, Choi et al. explored the feasibility of an HBV-derived 6-mer peptide Poly6 adjuvant combined with HBsAg used as a therapeutic vaccine to combat HBV infection in C57BL/6 mice or HBV transgenic mice [96]. Poly6 has strong anticancer capacity in tumor-implanted mice via the production of dendritic cells (DCs). The results revealed that Poly6 increased DC maturation and migration in a type-I interferon (IFN)-dependent mode in C57BL/6 mice [96]. They also found that HBsAg-specific cell-mediated immunity would be enhanced if Poly6 was added to alum to combine with HBsAg, hinting at its potential as an adjuvant of HBsAg-based vaccines. Additionally, vaccination with Poly6 combined with HBsAg showed a potent effect against HBV infection by eliciting humoral and cell-mediated immune responses via type-I IFN-dependent DC activation in HBV transgenic mice, for which the type I IFN receptorα-chain was knockout [96]. It also induced HBV-specific effector memory T cells. This study suggests that the Poly6 is a possible adjuvant for a therapeutic HBV vaccine to suppress viral infection significantly and reduce HBV antigens to a low level.

## 7. Perspectives and Challenges

### 7.1. Vaccine Administration and Application Can Be Diversified with Biotechnologies

Although vaccination is generally effective and advantageous in reducing or even eradicating infectious diseases, the development of vaccines for some diseases and emerging infections caused by pathogens still presents significant difficulties [97]. In particular, developing a vaccine for certain individuals with compromised immune systems or special medical conditions is challenging. Fortunately, some novel vaccines using biotechnologies, including vaccines designed for multiple/alternative administration routes, viral vector/nanoparticle vaccines, and mRNA vaccines, can provide practical strategies to overcome current challenges. The improvement of vaccine has also enormously promoted our understanding of vaccine immunology and comprehensive assessment for the future vaccine development [97]. Based on biotechnologies, it is feasible to extend the vaccine application from the prophylaxis of infectious diseases to noninfectious diseases (e.g., cancer) and from prevention to therapeutics of diseases. Potentially, some infectious diseases which have historically proven resistant to vaccination, like influenza and some rapidly emerging infectious diseases like COVID-19, could themselves be eradicated through vaccination based on novel biotechnologies. However, the unique challenges related to these emerging biotechnologies still need to be evaluated and overcome, for example, skin allergy to microneedles, local protection and unstable rates for inhalation, cytotoxicity of nanoparticles, and mutation of viral vectors (Table 1).

### 7.2. Vaccines Based on Biotechnologies Provide a Potential Cancer Therapy

The aging population will drive up medical care and pension payments because aging is a main risk factor for various chronic diseases. Vaccination provides a promising strategy against age-related diseases, such as Alzheimer’s disease, type II diabetes, hypertension, abdominal aortic aneurysm, atherosclerosis, osteoarthritis, fibrosis, and cancer, by targeting specific antigens and inducing immune responses [98]. The identification of disease-associated antigens and the induction of immune responses against these targets have facilitated the development of these vaccines. For the advancement of therapeutic potential, the next important step is to improve these vaccines and evaluate them in a clinical trial, accounting for their safety, efficacy, and long-term effects [98].

Therapeutic cancer vaccines are used to direct the immune system to combat cancer cells specifically. Because cancer cells are varied and different in individuals, it is difficult for vaccines to target cancer cells specifically. Therefore, the therapeutic vaccine must be designed individually and specifically; therefore, patients’ adaptation, economic situation, and response rate with respect to killing cancer cells should be evaluated. The technical development of these vaccines should particularly focus on the selection and delivery of specific antigens, the monitor of human post-vaccination responses, and the improvement of novel methods [99]. Neoantigens can provide a method by which to recognize cancer cells, but it is necessary to explore neoantigens using new biotechnologies and bioinformatic tools [99]. Moreover, it is important to induce tumor-specific T cells for immunotherapy and to extend patient survival. Therapeutic cancer vaccines have been proven to increase the number of T cells and their repertoire, but it is still difficult to elicit a significantly effective T cell population to achieve a minimum for the treatment of patients [99]. Overall, the unique challenges related to the development of therapeutic cancer vaccines include the requirement of personalized treatment (precision medicine), the difficulty of neoantigen exploration, and the insufficiency of effective T cells. Recent advancements in mass spectrometry, neoantigen prediction, and genetically engineered animal models can help us to overcome these challenges [99].

### 7.3. mRNA Vaccines Based on Biotechnologies Are Promising for Cancer Therapy

The development and approval of the COVID-19 mRNA vaccine based on nanoparticles has expedited the close investigation of the mRNA vaccine and encouraged more researchers to become involved in mRNA therapeutics. Currently, the studies related to mRNA vaccines for cancer therapy are rapidly increasing, and some substantial results in clinical trials have revealed that they can be used against various solid tumors [100]. mRNA antigens, adjuvants, and delivery vectors are the three major components of mRNA cancer vaccines, but they are either unstable or difficult to develop. The unique challenges related to the development of mRNA cancer vaccines include the instability of mRNA in vivo, the difficulties pertaining to the delivery vector (e.g., nanoparticle) and adjuvant discovery, and the complexity of component encapsulation. The engineering of these components can provide an approach to strengthen the therapeutic effects of mRNA cancer vaccines. For example, the stability and innate immunogenicity of mRNA can be improved by suitably modifying its structure, and the delivery efficiency in vivo can be elevated using appropriate mRNA delivery vectors [100]. Further strategies by which to improve the efficacy of mRNA cancer vaccines for eliminating tumors include the modulation of immunosuppressive tumor environment, the optimization of administration routes, the achievement of delivery vector targeting of intended cells or tissues, and the application of combination therapy [100]. If these challenges pertaining to the clinical use of mRNA cancer vaccines can be addressed and conquered, it is expected that mRNA vaccines will be rapidly applied for human cancer therapy in the near future.

## 8. Conclusions

Vaccination has been considered one of the most effective strategies for the prevention of infectious diseases. However, it is sometimes difficult to promote because of side effects, questions of efficacy, inconvenience, expense, and other issues causing vaccination hesitancy. Fortunately, novel administration routes, including microneedle and inhalation, can provide possible means of making vaccines more secure, comfortable, convenient, and economical. These two novel methods for vaccine delivery would contribute to the global tolerance and distribution of vaccines, especially in less-developed countries, general communities, and resource-limited areas. Vaccines based on novel vectors (e.g., viruses and nanoparticles) have the potential to improve safety and efficacy and make vaccination more tolerable and acceptable to most of the population. The extension of the vaccine application from the prophylaxis of infectious diseases to noninfectious diseases is another direction for vaccine development, particularly with respect to cancer prevention. High-cancer-risk individuals, such as those who are professionally exposed to oncoviruses or carcinogens or have a family history of cancer, are all suggested to receive vaccines. Importantly, therapeutic cancer vaccines, a promising approach within the rapidly growing field of immunotherapy, can provide cancer patients with an alternative therapeutic choice. The development of multiple administration routes, novel vectors, and extended usage based on biotechnologies has introduced a new era of vaccine administration and application. Based on biotechnologies such as the fabrication of microneedles; the preparation of inhalable dry powder; the development of delivery tools (e.g., virus, nanoparticle); the engineering of mRNA antigens, adjuvants, and delivery tools; and the encapsulation of mRNA and adjuvants in a delivery vector, novel vaccines are significantly different from traditional vaccines in terms of type, administration route, and application (Figure 2).

## Figures and Tables

**Figure 1 vaccines-12-01002-f001:**
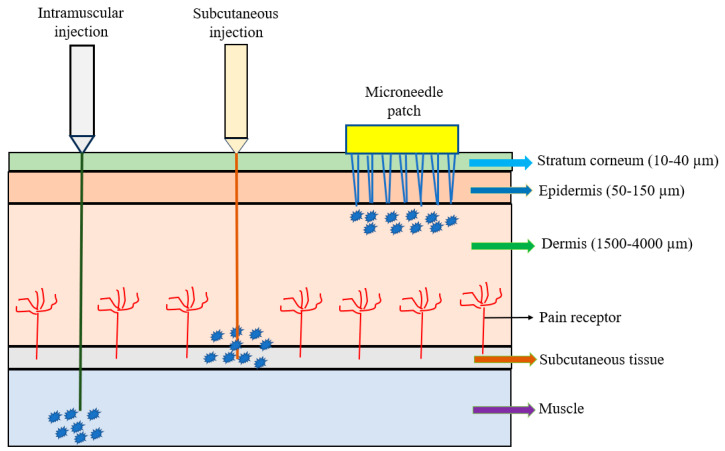
Comparison of intramuscular (IM) injection, subcutaneous (SC) injection, and microneedle patch.

**Figure 2 vaccines-12-01002-f002:**
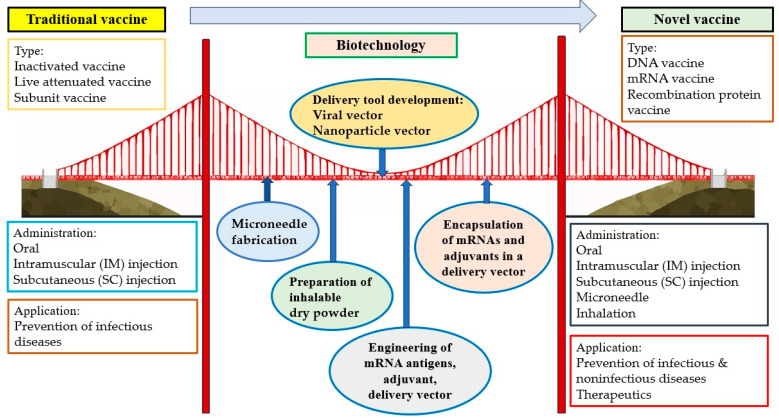
Building a bridge between traditional vaccines and novel vaccines based on biotechnologies.

**Table 1 vaccines-12-01002-t001:** Comparison of traditional and novel administration routes for vaccine delivery.

Administration Route	Traditional			Novel	
	Oral	Intramuscular	Subcutaneous	Microneedle	Inhalation
Advantages	Painless, self-administration, induction of mucosal immunity, herd immunity	Mildly contact with immune cells to induce immune responses	Long induction period by slow and sustained adsorption	Comfortable, minimal invasive delivery, self-administration, superior and rapid immunogenicity, longer induction period by slow and sustained adsorption, less reliance on cold-chain storage, dosage sparing effect [2,3,4,5]	Induction of triple immunity, including humoral, cellular, and mucosal immunity, intercepts pathogens at the first line when they invade, dosage sparing effect, self-administration [6,7,8,9]
Disadvantages, risk or limitations	First pass effect, environmental pollution caused by feces	Pain, inflammation, anxiety, infection, contamination, professionals, and cold-chain requirement	Pain, anxiety, inflammation, infection, contamination, lower immune responses, professionals and cold-chain requirement	Skin allergy, breakage of microneedle tip, foreign substances remaining in the body, thermostability must be monitored, sterilization is challenging [2,3,4,5]	Only suitable for respiratory or gastrointestinal infectious diseases, local protection, induction of immunotolerance, inhalation rate is unstable, induced immunity is difficult to evaluate [6,7,8,9]
Approved vaccine product example	Rotavirus vaccine: live attenuated vaccines Rotarix^®^ and RotaTeq^®^ [10], Poliovirus vaccine: Sabin, live attenuated oral polio vaccine (OPV) [11]	MMR (Measles, mumps, rubella), a live attenuated vaccine [12] Hexyon^®^ (Diphtheria, pertussis, tetanus, hepatitis B, poliomyelitis, and *Hemophilus influenzae* type b (Hib)), an inactivated vaccine [13] Poliovirus vaccine: Salk, an inactivated poliovirus vaccine (IPV) [14]	Bacillus Calmette-Guérin (BCG) vaccine, a live attenuated vaccine [15,16]	Influenza vaccine: Intanza^®^ and Fluzone^®^ [5]	Coronavirus Disease 2019 (COVID-19) vaccine: Convidecia Air^®^, an oral recombinant vaccine with adenovirus type 5 vector [17] iNCOVACC, an intranasal live attenuated vaccine [17,18]

**Table 2 vaccines-12-01002-t002:** Comparison of approved COVID-19 vaccines and Ebola vaccines using virus expression vectors.

Vaccine	COVID-19	Ebola Viruses
Viral vector serotype	Oxford–AstraZeneca vaccine: An adenovirus vector vaccine with modified chimpanzee adenovirus ChAdOx 1 [43]. Sputnik V vaccine: An adenovirus vector vaccine with human adenovirus serotype 26 (Ad26) for the first shot and serotype 5 for the second [44]. Janssen vaccine: An adenovirus vector vaccine with human adenovirus serotype 26 (Ad26) [45]. Convidecia™ vaccine: An adenovirus vector vaccine with human adenovirus serotype 5 (Ad5) [46].	Ervebo^®^ vaccine: A recombinant vesicular stomatitis virus (VSV)–Zaire Ebola virus (rVSV-ZEBOV); a recombinant and replication-competent viral vector vaccine, consisting of the rice-derived recombinant human serum albumin and a live attenuated recombinant VSV [47,48]. Zabdeno/Mvabea vaccine: An adenovirus vaccine with human adenovirus serotype 26(Ad26), expressing the glycoprotein of the Ebola virus. Mayinga variant [48,49]: This vaccine is delivered in two dosesl Zabdeno is given first and Mvabea is administered about 8 weeks later as the second dose.

**Table 3 vaccines-12-01002-t003:** Comparison of approved COVID-19 vaccines and influenza vaccines based on nanoparticles.

Vaccine	COVID-19	Influenza
nanoparticle type	Pfizer/BNT vaccine: mRNA encapsulated in lipid nanoparticles BNT162b2 mRNA [55]; the lipid components are the cationic lipid ALC-0315 combined with the phospholipid 1,2-distearoyl-sn-glycero-3- phosphocholine (DSPC), cholesterol, and a polyethylene glycol (PEG)–lipid [56]. Moderna vaccine: mRNA encapsulated in lipid nanoparticles mRNA-1273 [57]; the lipid components are SM-102 [Heptadecan-9-yl 8-((2-hydroxyethyl)(8-(nonyloxy) 8-oxooctyl)amino)octanoate)], PEG2000-DMG (1,2- dimyristoyl-rac-glycero-3-methoxypolyethylene glycol-2000), cholesterol, and DSPC (1,2-distearoyl-sn-glycero-3-phosphocholine) [56].	FluMos-v1 vaccine: A quadrivalent influenza nanoparticle vaccine offers long-lasting protection against multiple influenza virus strains and is composed of four strains of hemagglutinin trimer assembled around a pentamer core [58].

**Table 4 vaccines-12-01002-t004:** Comparison of different viral vectors for vaccine delivery.

Vector	Virus Expression Vector	Virus Nanoparticle	Virus-Like Particle
Characteristics	Live and infectious microorganisms but lack some viral genes	Live and infectious microorganism	Lifeless and noninfectious agent
Self assembly	Yes	Yes	Yes
Replication in hosts	Yes	Yes	No
Function for vaccine delivery	Delivery of DNA to encode specific antigens	Encapulation of antigens	Encapulation of antigens
Induced immunity	Both innate and adaptive	Both innate and adaptive	Both innate and adaptive

**Table 5 vaccines-12-01002-t005:** Comparison of viruses and nanoparticles used for vaccine development.

Vector	Virus	Nanoparticle
Characteristic	Living attenuated microorganisms, including virus expression vectors and virus particles Growth and replication in hosts	Specialized substance: non-living microorganisms including inorganic compounds, chemicals, liposomes, and virus-like particles (VLP), etc. Self-assembly without growth and replication in hosts
Advantage	Induction of both innate immunity and adaptive immunity Immunogenicity, immunogenic stability Highly efficient gene transduction and specific gene delivery to target cells Induction of potent immune responses An adjuvant is not needed Relatively low costs	The adverse reaction is usually less and milder because it is not alive and only induces adaptive immunity, except VLP, which induces both innate and adaptive immunity Protection of antigens and adjuvants Extension of antigen stability duration Enhancement of the antigen uptake Activation of dendritic cells Promotion of cross-presentation Effective encapsulation of mRNA
disadvantages, limitations or risks	Safety concern for viral reverse or spontaneous mutation Evaluation of the long-term duration of humoral and cellular responses is needed Understanding of the mechanism underlying antiviral immunity is critical, especially the following repeated dosing	Cytoxicity Evaluation of vaccine stability and delivery consistency is needed Development is difficult and expensive
Reference	[41,67,68,69,70]	[53,54,71,72,73]

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
