# Peer review of "Novel Administration Routes, Delivery Vectors, and Application of Vaccines Based on Biotechnologies: A Review"

_vaccines, 2024, doi:10.3390/vaccines12091002_

Round 1

Reviewer 1 Report

Comments and Suggestions for Authors

Rai et al. intended to review the current status of vaccine development using various biotechnology, which is a very hot research area. The difficulties of summarizing and writing such a big field are obvious. After reading this manuscript with interests, this reviewer feels the content is less attractive and the focus of this MS is not very clear. It is more like a commentary article or perspective and should not be treated as a "real" review paper. The following issues should be solved before it could be accepted in Vaccines.

1) The title is too big and not does not correctly reflect the contents of this paper. 

2) The whole ms needs significant improvement in the organization and structure. It is not clear why some specific biotechnologies and vaccines are selected for this review. We understand that various biotechnologies from biolabs are currently available and some of which have already been employed for producing commercial vaccines. What are the focuses? Tech or vaccines?

3) Since the design logic and clinical strategies have some variations, vaccines for pathogens and Tumors/cancers should be divided into different sections. 

4) The last section needs subtitles for each subsection.

Comments on the Quality of English Language

Moderate corrections.

Author Response

We greatly appreciate the comments and suggestions of reviewers and have revised the manuscript accordingly in tracked forms. Additionally, we have checked the vocabular and grammar carefully and asked an individual whose native language is not English to edit this manuscript.

Rai et al. intended to review the current status of vaccine development using various biotechnology, which is a very hot research area. The difficulties of summarizing and writing such a big field are obvious. After reading this manuscript with interests, this reviewer feels the content is less attractive and the focus of this MS is not very clear. It is more like a commentary article or perspective and should not be treated as a "real" review paper. The following issues should be solved before it could be accepted in Vaccines.

(1)The title is too big and not does not correctly reflect the contents of this paper. 

Ans: We have revised the title into “Novel Administration Routes, Delivery Vectors and Application of Vaccines based on Biotechnologies: A Review” (P.1, line 2-4)

(2) The whole ms needs significant improvement in the organization and structure. It is not clear why some specific biotechnologies and vaccines are selected for this review. We understand that various biotechnologies from biolabs are currently available and some of which have already been employed for producing commercial vaccines. What are the focuses? Tech or vaccines?

Ans: We have overall revised the manuscript to specify the biotechnologies and vaccines selected for this review. Our focus is novel vaccines based on biotechnologies.

(3) Since the design logic and clinical strategies have some variations, vaccines for pathogens and Tumors/cancers should be divided into different sections. 

Ans: We have divided “5. Vaccines Usd for Therapeutics” into “5. Vaccines Used for Pathogen Therapeutics and 6. Vaccines Used for Cancer Therapeutics”. (P.18, line 604~P.19, line 644)

(4) The last section needs subtitles for each subsection.

Ans: We have divided the last section “Perspectives and challenges” into 3 subsections. (P.20, line 686~P.22, line 759)

Reviewer 2 Report

Comments and Suggestions for Authors

The review titled "Development of Novel Vaccines Using Biotechnologies" explores the application of biotechnologies in vaccine research. While the topic holds potential, particularly in the areas of microneedles and inhaled vaccines, the manuscript faces significant issues that necessitate major revisions.

I recommend that the authors narrow the focus of the review to truly novel developments in vaccine delivery, specifically concentrating on microneedles and inhalation methods. The other vaccine administration routes and platforms discussed in the review, such as intramuscular, subcutaneous, and oral delivery, do not represent novel advancements. Claiming novelty in these areas could mislead readers, and the manuscript requires a complete restructuring to accurately reflect the state of the field.

The introduction is currently too brief and does not adequately outline the main points of the review. It needs to be expanded to better set the stage for the discussion that follows.

Regarding the section on vaccines based on administration routes, it is important to note that intramuscular, subcutaneous, oral, and even inhalation routes cannot be considered novel. The table in this section should be revised to include only genuinely novel delivery methods, and references should be added to support the claims made. Additionally, the authors should address the side effects of inhaled vaccines, which have been a significant issue in the past. This critical aspect was completely overlooked in the current manuscript and must be addressed to provide a comprehensive review.

https://www.frontiersin.org/journals/public-health/articles/10.3389/fpubh.2023.1047391/full

A well-rounded review would greatly benefit from the inclusion of illustrations, particularly to clarify the microneedle administration technique. Please consider adding visual representations to enhance the comprehensibility and appeal of this section.

Additionally, the section on microneedle administration currently lacks a detailed immunological explanation of its mechanism of action. I strongly recommend revising this part to provide readers with a clear understanding of how microneedles work at the immunological level. Incorporating illustrations here would further enhance the quality and clarity of the review.

It is also essential to clearly differentiate between inhaled and oral vaccines, as they are distinct in both their mechanisms and applications. This distinction needs to be thoroughly explained to avoid any confusion among readers.

Throughout the manuscript, many sections primarily list studies without effectively linking them to novel routes of administration or innovative vaccine design. To improve coherence and focus, the authors should revise these sections to ensure they convey a clear and concise message related to the core topic of novel vaccine development.

I recommend completely removing the section on cancer vaccines, as the content discussed (e.g., HPV, HBV, and TVEC) is neither novel nor likely to be of significant interest to the readers. The focus should remain on genuinely new advancements in vaccine technology.

Finally, the discussion on the respective challenges should be revised to reflect the context of novel technologies more specifically. The authors need to address the unique challenges associated with these emerging technologies to provide a comprehensive and insightful review.

 Thank you

Comments on the Quality of English Language

The English is poor. 

Author Response

We greatly appreciate the comments and suggestions of reviewers and have revised the manuscript accordingly in tracked forms. Additionally, we have checked the vocabular and grammar carefully and asked an individual whose native language is not English to edit this manuscript.

The review titled "Development of Novel Vaccines Using Biotechnologies" explores the application of biotechnologies in vaccine research. While the topic holds potential, particularly in the areas of microneedles and inhaled vaccines, the manuscript faces significant issues that necessitate major revisions.

Ans: We have added a figure (Figure 1) (P.4) and more description to strengthen the topics related to microneedles and inhalation vaccines. (P.4, line 81~84; P.7, line 185~192; P.7, line 198~P.8, line 233)

I recommend that the authors narrow the focus of the review to truly novel developments in vaccine delivery, specifically concentrating on microneedles and inhalation methods. The other vaccine administration routes and platforms discussed in the review, such as intramuscular, subcutaneous, and oral delivery, do not represent novel advancements. Claiming novelty in these areas could mislead readers, and the manuscript requires a complete restructuring to accurately reflect the state of the field.

Ans: We have revised the Table 1 to include oral, IM and SC as traditional routes and microneedles and inhaled as novel routes. (P.2, line 72). Additionally, we have changed to focusing on discussing microneedles and inhalation methods for vaccine delivery. (P.4, line 81~84; P.7, line 185~192; P.7, line 198~P.8, line 233)

The introduction is currently too brief and does not adequately outline the main points of the review. It needs to be expanded to better set the stage for the discussion that follows.

Ans: We have added one more section in the “Introduction”section to adequately outline the main points of the review to better set the stage for the following discussion. (P.2, line 44~54)

Regarding the section on vaccines based on administration routes, it is important to note that intramuscular, subcutaneous, oral, and even inhalation routes cannot be considered novel. The table in this section should be revised to include only genuinely novel delivery methods, and references should be added to support the claims made. Additionally, the authors should address the side effects of inhaled vaccines, which have been a significant issue in the past. This critical aspect was completely overlooked in the current manuscript and must be addressed to provide a comprehensive review.

https://www.frontiersin.org/journals/public-health/articles/10.3389/fpubh.2023.1047391/full

 Ans:

  • We have revised the Table 1 to include oral, IM and SC as traditional route and microneedles and inhaled as novel route. (P.2, line 72).
  • We have added more reference to support the claim in the Table 1, especially for microneedle and inhalation routes. (P.2, line 72)
  • We have added a statement including reference to address the side effects of inhaled vaccines. (P.7, line 198~207)

A well-rounded review would greatly benefit from the inclusion of illustrations, particularly to clarify the microneedle administration technique. Please consider adding visual representations to enhance the comprehensibility and appeal of this section.

Ans: We have added a figure (Figure 1) to illustrate microneedle administration technique. (P.4)

Additionally, the section on microneedle administration currently lacks a detailed immunological explanation of its mechanism of action. I strongly recommend revising this part to provide readers with a clear understanding of how microneedles work at the immunological level. Incorporating illustrations here would further enhance the quality and clarity of the review.

Ans: We have added a statement to explain the mechanism of action of microneedles (P.4, line 81~84) and a figure (Figure 1) to illustrate microneedle administration technique. (P.4)

It is also essential to clearly differentiate between inhaled and oral vaccines, as they are distinct in both their mechanisms and applications. This distinction needs to be thoroughly explained to avoid any confusion among readers.

Ans: We have revised the statement to clearly differentiate between inhaled and oral vaccines to show their difference in both mechanisms and applications. (P.7, line 208~P.8, line 233)

Throughout the manuscript, many sections primarily list studies without effectively linking them to novel routes of administration or innovative vaccine design. To improve coherence and focus, the authors should revise these sections to ensure they convey a clear and concise message related to the core topic of novel vaccine development.

Ans: We have revised the listed studies to effectively linking them to novel routes of administration or innovative vaccine design in every section, ensuring to convey a clear and concise message related to the core topic of novel vaccine development.

I recommend completely removing the section on cancer vaccines, as the content discussed (e.g., HPV, HBV, and TVEC) is neither novel nor likely to be of significant interest to the readers. The focus should remain on genuinely new advancements in vaccine technology.

Ans: We have removed the section on cancer vaccines as the content of HPV and HBV to focus on genuinely new advancements in vaccine technologies. (P.17, line 550~P.18, line 602)

Finally, the discussion on the respective challenges should be revised to reflect the context of novel technologies more specifically. The authors need to address the unique challenges associated with these emerging technologies to provide a comprehensive and insightful review.

Ans: We have revised the manuscript to reflect the context of novel technologies specifically and address the unique challenges associated with these emerging technologies. (P.20, line 703~706; P.21, line 733~738; P.21, line 747~P.22, line 759)

Reviewer 3 Report

Comments and Suggestions for Authors

This is a broad and interesting review; I enjoyed reading it. 

1. Page 17, 601: spelled COVID incorrectly

2. It would be useful to differentiate virus expression vectors, virus nanoparticles, and virus-like particles from each other. 

3. I particularly enjoyed reading about microneedle and inhalation delivery methods. Has this work also been performed for animal vaccines? It may be helpful to list a few. 

Comments on the Quality of English Language

The English is close to perfect. There are a few typos and inconsistencies, such as the first sentence: "Vaccines are biological preparations made of pathogenic microorganism to induce specific adaptive immune responses to against microbial infection like bacteria, viruses, etc." should be "Vaccines are biological preparations made of pathogenic microorganisms to induce specific adaptive immune responses to against microbial infection like bacteria, viruses, etc."

Author Response

We greatly appreciate the comments and suggestions of reviewers and have revised the manuscript accordingly in tracked forms. Additionally, we have checked the vocabular and grammar carefully and asked an individual whose native language is not English to edit this manuscript.

This is a broad and interesting review; I enjoyed reading it. 

  1. Page 17, 601: spelled COVID incorrectly.

Ans: We have revised the difference word correctly.

  1. It would be useful to differentiate virus expression vectors, virus nanoparticles, and virus-like particles from each other. 

Ans: We have added some statement and a table (Table 4) (P.13, line 434) to differentiate virus expression vectors, virus nanoparticles, and virus-like particles from each other. (P.10, line 324~326; P.13, line 418~433)

  1. I particularly enjoyed reading about microneedle and inhalation delivery methods. Has this work also been performed for animal vaccines? It may be helpful to list a few. 

Ans: We thanks for the reviewer’s recommendation. Microneedle and inhalation vaccines have been approved for human uses and more of these novel vaccines are being developed. Of course, this work has also been performed for animal vaccines. However, we do not want to include microneedle and inhalation delivery methods for animal vaccines in this manuscript, because we only explore vaccines for human use based on biotechnologies in this review. Anyway, we are pleased to list a few to the reviewer for reference as follows:

  • Edens C, Collins ML, Goodson JL, Rota PA, Prausnitz MR. A microneedle patch containing measles vaccine is immunogenic in non-human primates. 2015 Sep 8;33(37):4712-8. doi: 10.1016/j.vaccine.2015.02.074. Epub 2015 Mar 12.
  • Coupled Microneedle Approach. Arshad MS, Hussain S, Zafar S, Rana SJ, Ahmad N, Jalil NA, Ahmad Z. Improved Transdermal Delivery of Rabies Vaccine using Iontophoresis. Pharm Res. 2023 Aug;40(8):2039-2049. doi: 10.1007/s11095-023-03521-0. Epub 2023 Apr 25.
  • Calderon-Nieva D, Goonewardene KB, Gomis S, Foldvari M. Veterinary vaccine nanotechnology: pulmonary and nasal delivery in livestock animals. Drug Deliv Transl Res. 2017 Aug;7(4):558-570. doi: 10.1007/s13346-017-0400-9.
